# Environmental Impact on VOCs Emission of a Recycled Asphalt Mixture with a High Percentage of RAP

**DOI:** 10.3390/ma14040947

**Published:** 2021-02-17

**Authors:** Minghua Wei, Shaopeng Wu, Lei Zhu, Na Li, Chao Yang

**Affiliations:** 1School of Transportation, Wuhan University of Technology, Wuhan 430070, China; weiminghua@whut.edu.cn; 2State Key Laboratory of Silicate Materials for Architectures, Wuhan University of Technology, Wuhan 430070, China; hbyangc@whut.edu.cn; 3Technology Center of Dongfeng Automobile Group Co. Ltd., Wuhan 430056, China; zhulei@dfmc.com.cn

**Keywords:** high content RAP, recycled asphalt mixture, servicing performance, VOCs emission

## Abstract

Recycling of reclaimed asphalt pavement (RAP) has gradually emerged as a hot topic in the current research of building materials. Manufacturing the recycled asphalt mixture with a high RAP content still remains a major challenge due to the problem of inferior water damage resistance and fatigue cracking resistance. The aim of this study is to evaluate the servicing performance of recycled asphalt mixture with high percentage of RAP and assess its environmental impact on volatile organic compounds (VOCs) emission. To further explore the intrinsic factor on the performance of a recycled asphalt mixture, the mixed asphalt binder with a different content of extracted RAP binder was firstly investigated. The results show that the larger the content of RAP binder, the lower the mechanical indexes and VOCs emission. There exists an internal relationship between the chemical composition and VOCs release behavior with the variation of the recycled asphalt binder content. Based on the results of mixed recycled asphalt binder, the road servicing performance of a rather high utilization of RAP (i.e., 50%, 60%, and 70%) was assessed. It was found that the reuse of RAP aggregates has little influence on the volume performance of recycled asphalt mixture, and servicing performances still meet the construction requirements in spite of a descending trend. Moreover, a significant reduction effect on VOCs emission was found in the mixing stage of recycled asphalt mixture, indicating that the VOCs emission can be decreased by 94.82% when the content of RAP aggregates increases to 70%. The recycling of RAP with a high content contributes to the sustainable development of road engineering and the construction of green pavements.

## 1. Introduction

With the rapid development of the economy and society, China’s pavement construction has also undergone earth shaking changes. By the end of 2019, the total mileage of national highways reached 5.0125 million km, of which the total mileage of highway maintenance occupied 98.8% [1]. Roads infrastructure has changed from being construction oriented to focusing on both construction and maintenance. Due to advantages of smooth surfaces, comfortable driving conditions and convenient maintenance, asphalt pavement has become the main form of pavement in China, accounting for more than 80% of the total pavement area [2]. Milling the original existing pavement surfaces and then paving the new asphalt pavement is the most common form of maintenance treatment for asphalt pavement, producing a lot of waste asphalt mixture which causes environmental pollution. Considering that asphalt pavement construction and usage is an energy and resource intensive and high emissions industry, the reusing of Reclaimed Asphalt Pavement (RAP) can be treated as a prospective technology for reducing the consumption of natural resources and bringing benefits to the environment [3,4,5]. It has been reported that the cost of asphalt mixture production can be reduced from 70 US dollars per ton to 50 US dollars per ton by adding 50% RAP, which can even be reduced by 50–70% when the reuse of RAP reaches 100%. Meanwhile, a saving of 35% of CO_2_ emissions can be realized with a fully 100% RAP asphalt mixture substituting for the traditional asphalt mixture [6]. Therefore, it is useful to conduct research on recycling technology of asphalt pavement which is in line with the development requirements of green and sustainable transportation in the period of large-scale maintenance.

Several studies have tried to prove the possibility of the substitution of RAP in asphalt mixture. Izaks found that there were no significant differences in mechanical behavior between the RAP used and conventional asphalt mixture. Although the addition of RAP can decrease the fatigue resistance of asphalt mixture, it still met the necessary performance requirements [7]. Widyatmoko evaluated the mechanical properties of asphalt mixtures with different proportions such as 10%, 30% and 50%. The reuse of RAP had a similar quality to conventional mixtures [8]. Pradyumna confirmed that a better performance existed in the sample with 20% RAP than in the conventional mixture under similar conditions [9]. In spite of the availability of RAP applied in asphalt concrete, the utilization rate of RAP in hot mix recycled asphalt mixture is generally set as 10–25%, since the application technology is relatively mature [10,11,12]. It was found that the increase in RAP content (0–100%) significantly influenced the properties of recycled asphalt mixtures [13]. The annual production of RAP in China can reach up to 220 million tons, for which the relative low utilization rate is far from enough to meet the needs of waste recycling and landfill space reduction [14]. It is a urgent demand and great challenge to design the asphalt mixtures with a high RAP content and evaluate their performance in the result of an increment in the stiffness of aged asphalt binder [15].

Great progress has been made in the research of recycled asphalt mixture with a high content of RAP, mainly focusing on the improvement of processing technology, the addition of rejuvenating agents and the assessment of mechanical performance [16,17,18,19,20,21]. It is practicable to apply 50% or even more RAP under an optimal condition to meet the requirement of the long-term performance of asphalt pavement in spite of more rutting and crack diseases occurring [22,23]. However, research rarely revealed the environmental impact induced by the application of recycled asphalt mixtures, especially on atmospheric pollution emissions. Asphalt, as the most important part of an asphalt mixture, had been confirmed to cause emissions of volatile organic components (VOCs) during the heating process, most of which are hazardous to ozone formation, SOA yields and human health [24,25,26]. The asphalt binder is vulnerable to oxygen and solar radiation during long-term periods with variations of chemical structure changes. The VOCs emission behavior is sensitive to the asphalt binder and processing regime and it is still unclear how VOCs are affected by the use of recycled asphalt mixture.

Considering the increasingly severe atmospheric pollution, exploration of VOCs emission during asphalt pavement construction with high percentages of RAP is a topic worthy of attention. The main objective of this study is to investigate and quantify the VOCs emission with a high percentage of RAP in a recycled asphalt mixture. Firstly, the internal relationship between chemical composition and VOCs release behavior was revealed with the variation of recycled asphalt binder content. Furthermore, the VOCs release from recycled asphalt mixtures with different contents in the mixing process was monitored to combine them in a practical situation and the environmental impact of doing this was further evaluated. This study also explores the potential VOCs reduction effect of the application with a recycled asphalt mixture.

## 2. Materials and Methods

### 2.1. Materials

The RAP materials in this study were obtained from the upper and middle surface layer of the Hanshi section of the Wu-Huang expressway. The particle size of RAP is generally 0–20 mm (as shown in Figure 1). Steel slag from Baotou Iron and Steel Co. Ltd. (Baotou, China) and the SBS (Styrene–butadiene-styrene Copolymer) modified asphalt from Inner Mongolia Luda Asphalt Co. Ltd. (Huhehot, China) were used as the coarse aggregate and binder in this research, respectively. 

To further investigate the intrinsic factor on the performance of recycled asphalt mixture, the RAP binder was obtained by the centrifugal extraction method. The fundamental properties of the SBS asphalt binder and RAP binder are listed in Table 1. It can be seen that after long-term service, the asphalt in the recycled asphalt mixture has experienced a significant aging period, of which the penetration is only 45 mm, while the penetration requirement of SBS modified asphalt is higher than 60 mm (0.1 mm). The ductility at 5 °C also presents a significant decrease to only 18.7 cm. At the same time, the softening point significantly increases to 73 °C, meaning the aged asphalt becomes brittle and has an enhanced temperature sensitivity. Hence, the asphalt rejuvenator agent as RA102 from Jiangsu Subote New Material Co. Ltd. (Nanjing, China) were chosen to improve the performance of the mixed recycled asphalt binder and the optimum content was set to be 6% of asphalt quality based on comprehensive performance indexes.

### 2.2. Sample Collection

The asphalt VOCs belong to the unorganized emission category, which is vulnerable to the external environment and is not conducive to sample collection. The VOCs emission from the recycled asphalt binder was collected in a self-designed equipment with a sealed reaction vessel (as shown in Figure 2), which has been introduced in our previous work [25]. Before starting to collect gas samples, it was necessary to use the high purity nitrogen for the replacement of a gas sampler bag that fills with gas and then lets it go, and repeating these steps five times. When the last replacement was finished, the sampling pump was used to empty the gas in the sampling bag, which was waiting for the sample collection. The virgin SBS asphalt binder and extracted RAP binder (40%, 50%, 60%, 70% and 80% by the total weight of asphalt binder mixture, respectively) were mixed at 160 °C. They began to collect a released gas sample when the mixed asphalt binder was stirred at a constant rate of 300 r/min for 10 min to ensure the homogeneity of the mixture. An atmospheric sampler pump connected with an atmospheric sampler bag was applied to collect the generated gas, of which the sampling rate set as 500 mL/min and the collection time was set as 4 min. To avoid the influence of large size particles, a filter membrane was assembled in the gas inlet.

In order to fit the actual production process, this paper mainly studied the VOCs emission behavior in the mixing process of asphalt mixture owing to the relatively high temperature of asphalt mixture mixing in the whole construction process. Due to the lack of research on VOCs emission of asphalt mixtures, there is no suitable gas generation device to achieve this. Based on the mixing experiment of asphalt mixture in our laboratory, this paper tried to restore the mixing process as much as possible. By designing a baffle with collecting holes above the mixing pot (as shown in Figure 3), the VOCs emission gas was collected in a similar closed environment. 

Based on the performance of different RAP binder mixture, the dosage of RAP aggregate was set as 50%, 60% and 70%, respectively. Meanwhile, the conventional asphalt mixture without RAP was also prepared as the reference sample. The commonly used asphalt mixture (AC-13) was designated in this research and the aggregate grading curves is shown in Figure 4. According to the Marshall Test, the optimal asphalt content was 4.8%, 4.9% and 5.3% corresponding to 50%, 60% and 70% additions of RAP. After the mixing process was finished, we quickly covered the baffle when the mixing pot descended to avoid VOCs escaping directly into the surrounding environment, thus affecting the final test results. At this time, the gas sampling pump was turned on at the sampling rate to 500 mL/min for 4 min to collect the released VOCs. 

### 2.3. Rheological Properties Test

The rheological property of asphalt binder was tested by dynamic shear rheometer (DSR) (MCR101, Anton Paar, Graz, Austria). A temperature sweep was performed firstly in a range from 30 °C to 80 °C and was then fixed at 10 rad/s. Then a frequency sweep was performed in the same temperature range to the temperature sweep. Twelve temperature levels were obtained at temperature intervals of 5 °C. After reaching the target temperature, the samples were scanned in a range of 0.1–100 Hz of frequency. The strain was 2.0%. 

### 2.4. Fourier Transform Infrared Spectroscopy (FTIR) Test

The microstructural characteristics of the asphalt binder were tested by a Fourier transform infrared spectroscopy (Nicolet6700, Thermo Nicolet, MA, USA). The asphalt film was used for the test, which was prepared by dissolving 0.1 g of a sample in 2 mL of CS_2_ and then dropping the solution onto KBr wafer until it volatilized completely. The scan range was set from 4000 cm^−1^ to 400 cm^−1^ with a resolution of 4 cm^−1^. To quantify the asphalt chemical structure properties, four functional group indexes were selected to compare the structural differences of asphalt sample, which are the carbonyl functional group index (1700 cm^−1^) and sulfoxide functional group index (1030 cm^−1^) reflecting the aging degree, the aliphatic functional group index (1376 cm^−1^ and 1456 cm^−1^) reflecting the relative content of aliphatic compounds such as long-chain alkanes and finally the aromatic functional group index (1600 cm^−1^) reflecting the relative content of benzene ring substitutes. The calculation formulas of the four functional group indexes are shown in Equations (1)–(4) as follows [27,28,29]:(1)$$I_{co}=\frac{A_{1700^{-1}}}{A_{2000-600^{-1}}}$$
(2)$$I_{so}=\frac{A_{1030^{-1}}}{A_{2000-600^{-1}}}$$
(3)$$I_{cc}=\frac{A_{1376^{-1}}+A_{1456^{-1}}}{A_{2000-600^{-1}}}$$
(4)$$I_{AR}=\frac{A_{1600^{-1}}}{A_{2000-600^{-1}}}$$
where: *A*_1700_^−1^: Area of the carbonyl centered around 1700 cm^−1^, *A*_1030_^−1^: Area of sulfoxide band centered around 1030 cm^−1^, *A*_1376_^−1^: Area of saturated C-H band centered around 1376 cm^−1^, *A*_1456_^−1^: Area of asymmetric stretching vibration of C-H band centered around 1456 cm^−1^, *A*_1600_^−1^: Area of benzene ring centered around 1600 cm^−1^, *A*_2000–600_^−1^: Area of spectra ranged from 2000–600 cm^−1^.

### 2.5. Servicing Performance Test

The mechanical properties of recycled asphalt mixtures were characterized with the Marshall Stability test, tensile strength ratio (TSR), rutting test and semi-circular bending test at low temperatures based on the instructions of, “Standard Test Methods of Bitumen and Bituminous Mixtures for Highway Engineering [30]”. 

### 2.6. Asphalt VOCs Test

The qualitative and quantitative determination of VOCs emission were conducted with the GC-MS technique (Agilent GC7820A/MS5977E, Santa Clara, CA, USA). The VOCs were firstly leaded to a pre-concentration system (Tianhong, TH-300B, Wuhan, China) with a self-contained pump and frozen at −150 °C. The species were separated due to the difference in desorption ability of chemical components upon increasing the temperature to 180 °C and holding for 5 min. In the whole process, helium gas (≥99.999%) was used as the carrier and the heating rate was kept as 5 °C/min. The separated species were transported to a MS detector to identify the detailed composition over the range of m/z 35–200. A standard gas calibrator Restek, PA, USA) depended on the requirement of Photochemical Assessment Monitoring Stations (PAMS) and EPA TO-15 was used to quantify the VOCs amount.

## 3. Results and Discussion

### 3.1. Effect of Mixed Recycled Asphalt Binder

#### 3.1.1. Performance Evaluation 

In this study, SBS asphalt binder, recycled asphalt binder and mixed asphalt binder with different proportions (40%, 50%, 60%, 70% and 80%) were selected as the research objects, which were named as SBS, R, R-40%, R-50%, R-60%, R-70% and R-80%, respectively. The complex modulus of the mixed recycled asphalt binder with different ratios is shown in Figure 5. It can be seen that with the increase of the temperature, the |G*| values of all the asphalt binders presented a downward trend because of the softer asphalt binder at high temperatures. The virgin asphalt binder exhibited the smallest |G*| value, while with the increase of the content of the RAP binder, the |G*| value of the mixed recycled asphalt binder increased gradually, meaning the mixed asphalt gradually hardened. Especially when the content of RAP binder increases to 80%, the |G*| value of the mixed asphalt is close to that of the entire aged asphalt binder. 

The phase angle δ of each mixed asphalt binder is shown in Figure 6. The variation trend of the phase angle of mixed asphalt binder with the temperature can be seen to be the opposite to that of the |G*| value. The phase angle of the virgin asphalt is the largest, the mixed asphalt is the second and the extracted RAP asphalt is the lowest. With the increase of the RAP binder, the phase angle of mixed asphalt binder decreases gradually. The phase angle of the recycled asphalt binder with a lower content such as 40% and 50% is close to that of virgin asphalt. If the addition of RAP binder is less than 70%, then the effect on the phase angle is not significant. However, when the RAP binder content reaches 80%, the phase angle of the mixed asphalt binder decreases obviously and gradually approaches that of the extracted RAP binder.

#### 3.1.2. Chemical Structure Analysis

FTIR analysis technology is a conventional method for organic species analysis, and the infrared spectrum of asphalt mainly consists of two parts: the frequency region of organic functional groups and the fingerprint region of asphalt. The frequency range of organic functional groups is 4000–1300 cm^−1^, which mainly involves the vibration, stretching and absorption of different chemical components. This is the major focus for studying the chemical structure of asphalt binder. By comparing the infrared spectra of SBS and R (as shown in Figure 7), it was found that the infrared spectra of the two kinds of asphalt binder are similar in both the shape and position of characteristic absorption peaks of functional groups, but the areas of absorption peaks are quite different. The absorption peaks of the RAP binder at 1030 cm^−1^ and 1700 cm^−1^ are obviously enhanced, corresponding to the vibration of sulfoxide group (S=O) and carbonyl group (C=O) respectively, which indicates that the recycled asphalt binder has obvious aging.

Further comparing the infrared spectra of recycled asphalt binder with different mixing ratios (as shown in Figure 8), it was found that with the increase of the RAP binder ratio, the intensity of absorption peaks corresponding to 1376 cm^−1^, 1465 cm^−1^, 1600 cm^−1^, 2857 cm^−1^ and 2930 cm^−1^ gradually weakens. These peaks belong to the umbrella vibration of methyl-CH_3_^−^, shear vibration of methylene-CH_2_^−^, respiratory vibration of benzene ring and methylene ring, the symmetric stretching vibration of C-H and the asymmetric stretching vibration of C-H in methylene, respectively. This phenomenon reveals the fading of the vibration of aliphatic and aromatic functional groups. However, the absorption peak of 1700 cm^−1^ gradually increases, indicating that the vibration of oxygen-containing organic functional groups is enhanced with more obvious aging degree conditions.

#### 3.1.3. VOCs Emission Assessment

The total VOCs emission concentration of different samples is depicted in Figure 9. The results show that SBS asphalt binder releases the largest number of VOCs at 11.15 mg/m^3^, while the recycled asphalt binder R released the least VOCs with a concentration of 6.80 mg/m^3^, which was reduced by 39%. When the two kinds of asphalt binder were fully mixed in different proportions, the VOCs emission was basically between that of the pure SBS asphalt binder and the recycled asphalt binder. The VOCs released from R-40% asphalt binder were 28% less than that from the SBS asphalt binder and the recycled asphalt binder had an obvious effect on reducing VOCs emission. Then, with the increase of the RAP binder content, the emission of VOCs decreased gradually, and the trend tended to be gentle. When the content of the RAP binder reached 70%, the emission of VOCs was almost the same as that of the pure RAP binder and the emission reduction effect was no longer obviously enhanced. 

Considering the complex chemical composition and very small fraction of each species of asphalt VOCs, it is a great challenge to analyze a single substance. Hence, four groups of chemical substances: aliphatic hydrocarbons (ALH), aromatic hydrocarbons (ARH), oxygen containing hydrocarbon derivatives (O-HYD) and halogen containing hydrocarbon derivatives (H-HYD) were clarified depending on the characteristics of the chemical group. Figure 10 shows the distribution of VOCs category in the mixed recycled asphalt binder. Among them, the content of ALH released from the SBS asphalt binder was the highest, accounting for 55% of the total VOCs detected, followed by O-HYD and ARH while the content of H-HYD was the minority. However, in the VOCs released from the recycled asphalt binder R, the proportion of O-HYD increased significantly to 49.02%, which was 25% higher than that in the SBS asphalt binder. At the same time, the content of ALH decreased to 35.38%, indicating that ALH was oxidized and gradually converted into O-HYD. ARH and H-HYD presented a downward trend, but compared with the other two components, the variation was not significant. It was found that the contents of ALH, ARH and H-HYD decreased with the increase of the RAP binder ratio, but the content of O-HYD increased gradually. When the ratio of RAP binder reached 70%, the effect was not obvious. The influence of RAP binder was mainly on the release of ALH and O-HYD, which accounted for more than 80% of the total VOCs release.

In this paper, the intrinsic factor on VOCs emission from the different mixed recycled asphalt binder was further discussed by characterizing the chemical molecular structure with infrared spectrum analysis. The functional group index was calculated with each main characteristic absorption peak area depending on the composition distribution of VOCs in the asphalt binder mentioned above, such as the aliphatic functional groups corresponding to 1376 cm^−1^ and 1456 cm^−1^, aromatic functional groups corresponding to 1600 cm ^−1^, and carbonyl (C=O) functional groups corresponding to 1700 cm^−1^. The comparison between the distribution of VOCs components and variation of the corresponding functional group index is shown in Figure 11. With the increase of the RAP content in mixed recycled asphalt binders, the content of ALH and ARH decreased with the reduction of the aliphatic and aromatic functional group index, and the proportion of O-HYD increased with the enhancement of the carbonyl (C=O) functional group index. However, the change rate of ALH and O-HYD was significantly higher than that of the aliphatic and carbonyl (C=O) functional group index, which indicates that on the one hand, the chemical structure of recycled asphalt binder had an impact on the composition and distribution of asphalt VOCs; on the other hand, during the heating process, the ALH volatilized from asphalt were further oxidized under the reaction with temperature and oxygen, accelerating the formation of oxygenated compounds.

### 3.2. Effect of Recycled Asphalt Mixture with High Percentage of RAP

#### 3.2.1. Performance Evaluation

According to the rheological properties of mixed recycled asphalt binder, the index seriously deteriorates when the RAP binder content is 80%. Therefore, the RAP aggregate content in the recycled asphalt mixture was selected to be 50%, 60% and 70% under comprehensive consideration. The servicing performance of recycled asphalt mixture is shown in Table 2. It was found that the void ratio of recycled asphalt mixtures ranged from 3.91–4.08%, meeting with the index that should be below 5% in the current asphalt pavement construction technical specifications. The freeze-thaw splitting test and immersion Marshall stability test were used to explore the water stability of recycled asphalt mixture. It was found that with the increase of the RAP aggregate content, the residual stability, tensile strength ratio and low-temperature three-point bending tensile strength demonstrated a downward trend, indicating a negative impact on the water stability and low temperature crack resistance. Although the performance deteriorated to some extent with the addition of RAP, the performance could still meet the requirements. As a kind of viscoelastic material, it is necessary to note that the high temperature stability of recycled asphalt mixture under the substantial addition of RAP due to the viscosity of asphalt will gradually decrease with the increase of the temperature. The dynamic stability in the rutting test of all of the recycled asphalt mixtures is much higher than the 2400 frequency/mm as required. The increase of RAP in the recycled asphalt mixture can enhance the rutting performance. The index reaches 4701 frequency/mm when the RAP content is 70%, which is enhanced by 11.2% compared with that of 50% RAP. 

#### 3.2.2. VOCs Emission Assessment

In the practical mixing process of asphalt mixture, the release condition of VOCs in asphalt mixture is quite different from that in asphalt binder, due to the contact effect of aggregate, mineral powder and asphalt binder, as well as the different mixing process. In order to fit the actual situation, this paper continues to study the VOCs emission behavior of recycled asphalt mixture with a high percentage of RAP during the mixing stage, as shown in Figure 12. The results show that the addition of RAP can significantly reduce the emission of VOCs in the mixing process of asphalt mixture. When the amount of RAP is 50%, the emission of VOCs decreases by 50.30% compared with the conventional asphalt mixture without RAP. Moreover, when the content of RAP increases to 70%, the amounts of VOCs released in the mixing process of asphalt mixture is very low and the emission reduction rate reaches 94.82%.

By analyzing the concentration variation of VOCs groups in different kinds of asphalt mixture in the mixing stage (as shown in Figure 13), it was found that the content of ALH is the highest, accounting for almost half of the total VOCs released. With the addition of RAP, the contents of four categories decrease obviously. When the RAP content is 50%, the decrease of ALH is the largest with a decrease of 51.75%, followed by ARH and O-HYD with decreases of 50.52% and 42.29%, respectively, compared with the conventional asphalt mixture. The addition of RAP aggregate has the greatest influence on the reduction of O-HYD, followed by ALH and ARH. As the overall emissions of H-HYD are small, the impact is not as obvious as for other substances. The influence of RAP on the VOCs emission of asphalt mixture mainly exists in two aspects: on the one hand, the light components as ALH in the RAP aggregate has been volatilized or oxidized after aging in service, which is the majority composition in asphalt VOCs; on the other hand, the addition of the RAP aggregate is beneficial to reduce the amount of new asphalt, which helps to reduce VOCs emission.

## 4. Conclusions

This study explored the availability of manufacturing recycled asphalt mixtures with high percentages of RAP and evaluated their environmental impact on VOCs emission. Chemical structures, rheological properties and servicing performance (Marshall Stability, tensile strength ratio, rutting dynamic stability and bending test at low temperatures) were considered as the evaluating indicators. The VOCs emission behavior from both mixing stages of recycled asphalt binder and asphalt mixtures were further investigated. The following conclusions can be summarized:(1)The rheological properties of the mixed recycled asphalt binder are found to be between those of the virgin SBS asphalt and RAP binder and they gradually deteriorated with the increase of content of the RAP binder. Especially when the RAP binder content reaches 80%, the relative performance approaches that of the extracted RAP binder.(2)The addition of a high percentage of RAP aggregate will not affect the volume performance of recycled asphalt mixture. Although the water damage resistance, high temperature stability, low temperature cracking resistance and fatigue cracking resistance demonstrate a downward trend with the increase of RAP aggregate, the road performance still meets the construction requirements.(3)In addition to the change of the relative content of specific functional groups, the addition of RAP will not introduce new substances. There exists an internal relationship between chemical composition and VOCs release behavior with the variation of recycled asphalt binder content. The increase of the RAP binder mainly affects the release of ALH and O-HYD, which account for more than 80% of the total VOCs release.(4)The recycled asphalt mixture with a high RAP content has an obvious emission reduction effect on VOCs. When the content of RAP aggregate is 50%, the VOCs emission can be decreased by 50.30% during the mixing stage compared with the asphalt mixture without RAP. The VOCs emission reduction can be up to 94.82% if the content of RAP aggregates increases to 70%. Maximizing the utilization of RAP can not only make the use of raw materials sustainable, but also is an effective method to enhance environmental protection.

## Figures and Tables

**Figure 1 materials-14-00947-f001:**
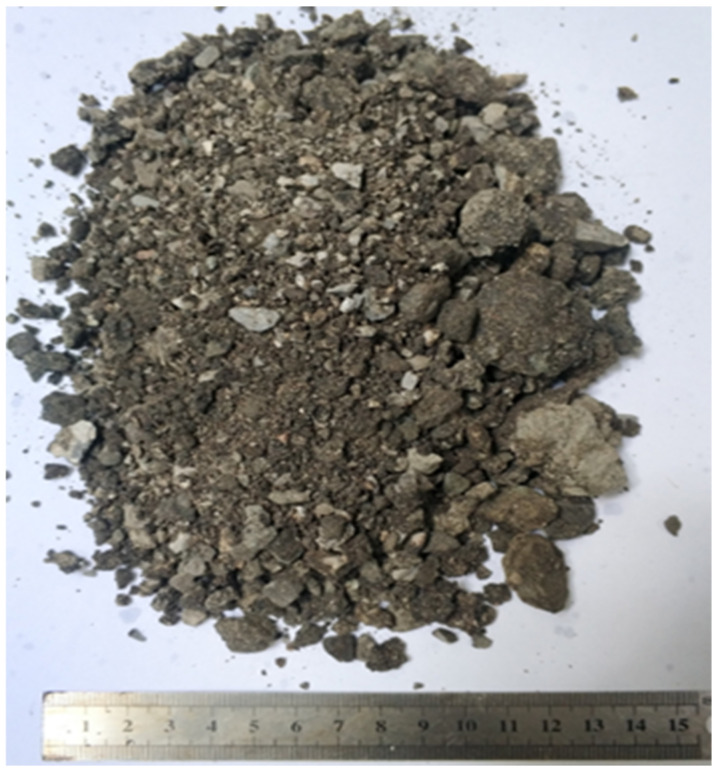
Reclaimed asphalt pavement (RAP) of the Wu-Huang Expressway.

**Figure 2 materials-14-00947-f002:**
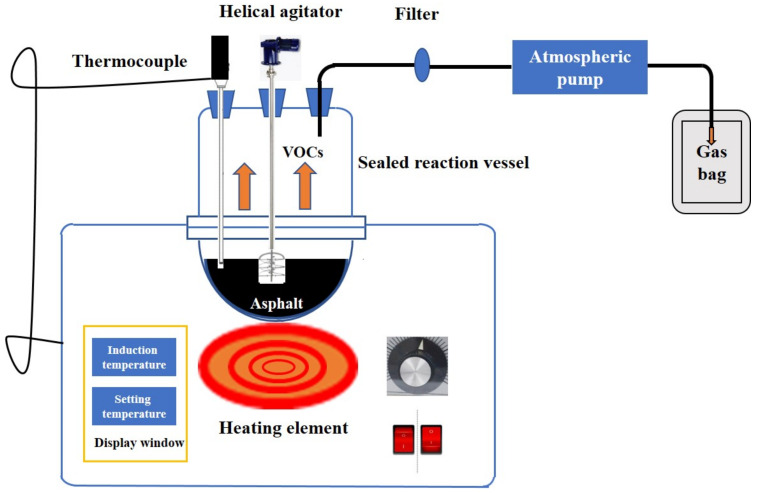
Schematic of equipment for mixed recycled asphalt binder volatile organic compounds (VOCs) emission collection (Adapted with permission from ref. [25]. 2021 Elsevier.).

**Figure 3 materials-14-00947-f003:**
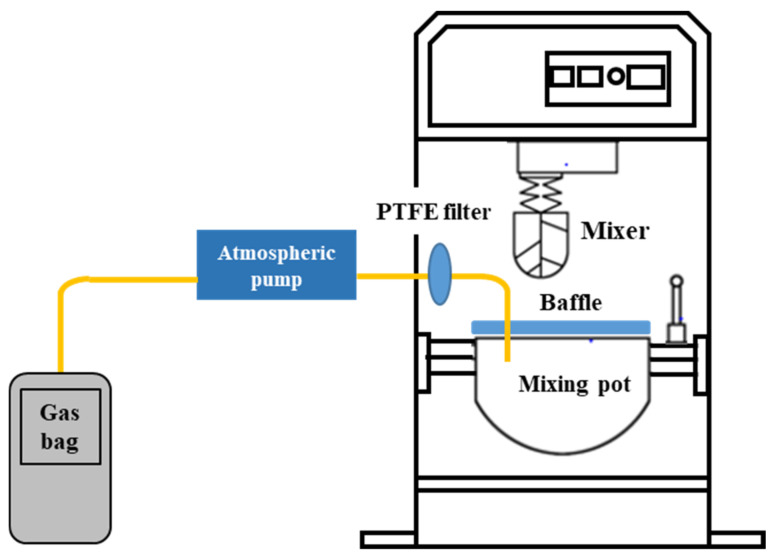
Schematic of equipment for recycled asphalt mixture VOCs emission collection.

**Figure 4 materials-14-00947-f004:**
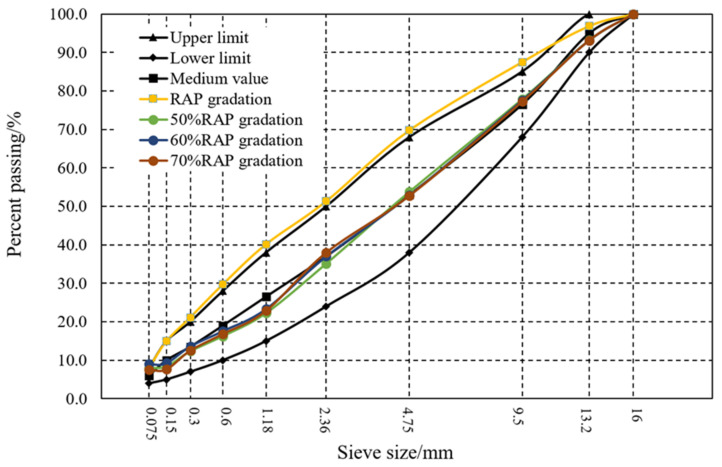
Grading curve of the AC-13 asphalt mixture with a different RAP ratio.

**Figure 5 materials-14-00947-f005:**
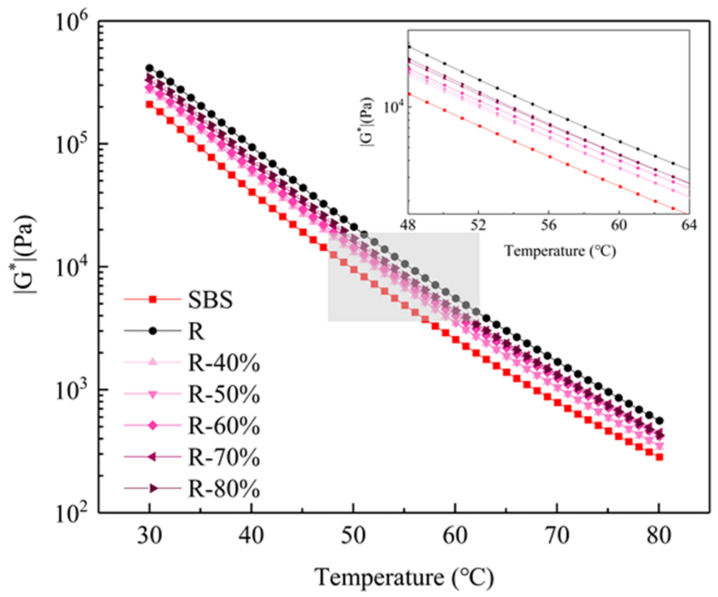
Complex modulus of the mixed recycled asphalt binder.

**Figure 6 materials-14-00947-f006:**
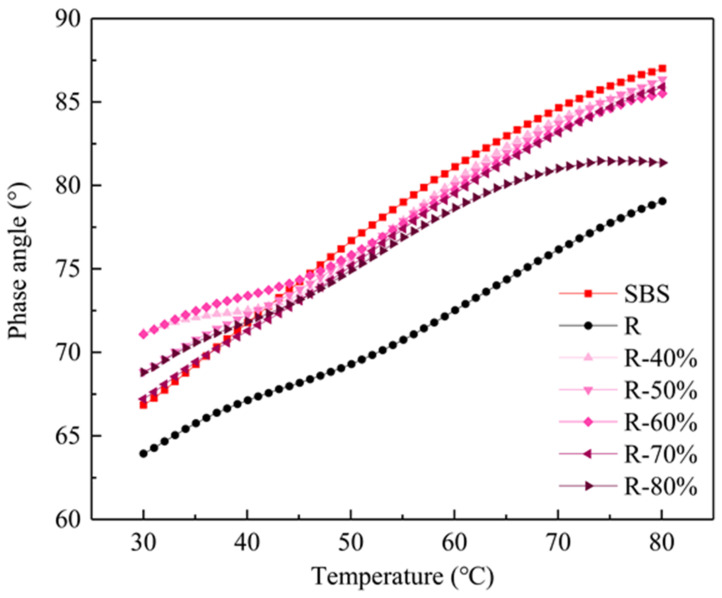
The phase angle of the mixed recycled asphalt binder.

**Figure 7 materials-14-00947-f007:**
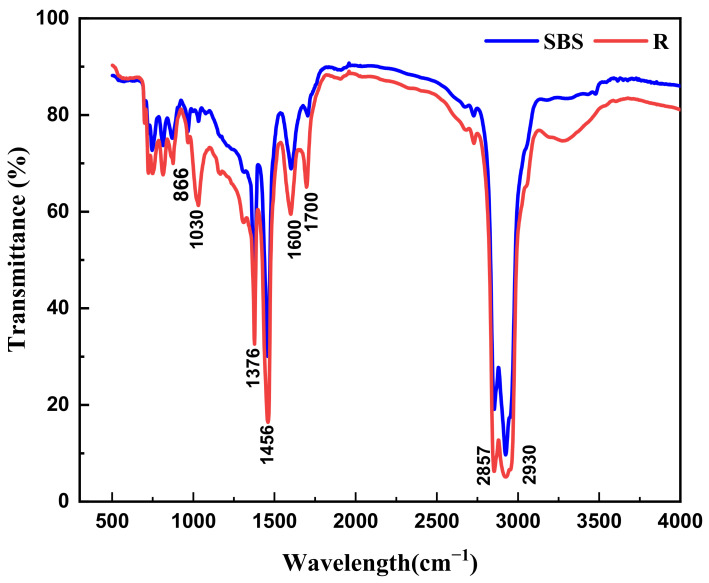
Infrared spectrum of the SBS asphalt binder and RAP binder.

**Figure 8 materials-14-00947-f008:**
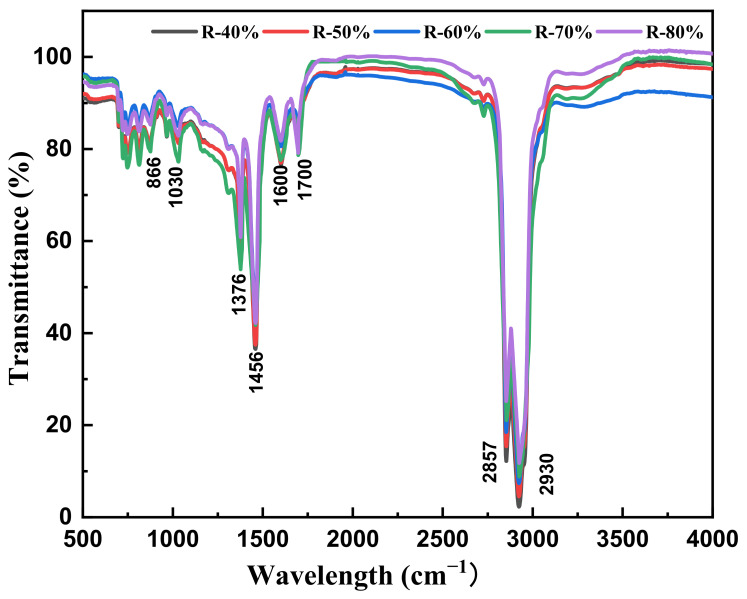
Infrared spectrum of recycled asphalt binder with a different mixing ratio.

**Figure 9 materials-14-00947-f009:**
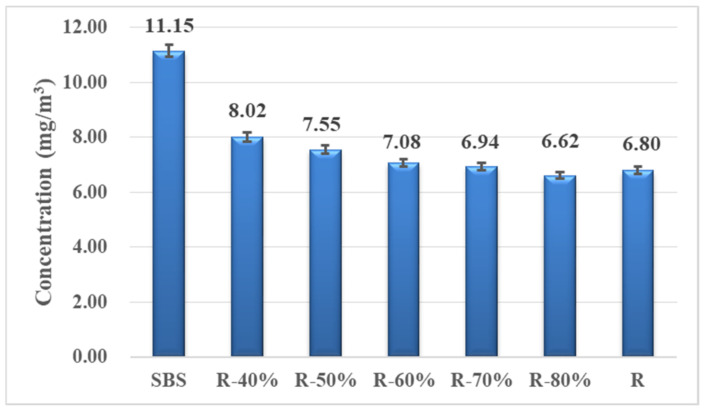
VOCs emission concentration of recycled asphalt binder with a different mixing ratio.

**Figure 10 materials-14-00947-f010:**
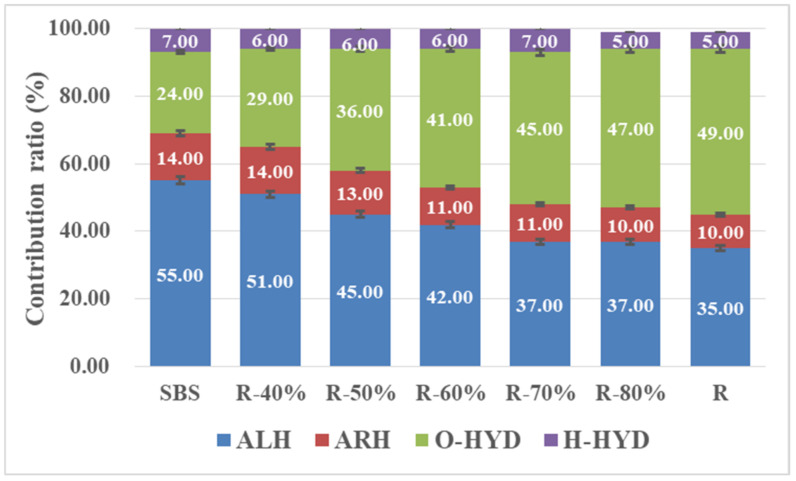
Composition distribution of VOCs in recycled asphalt binder with different mixing ratio.

**Figure 11 materials-14-00947-f011:**
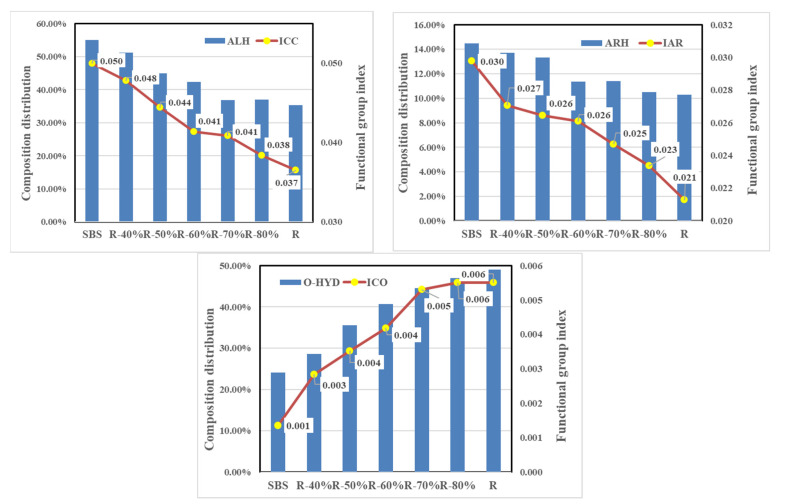
Comparison of between the distribution of VOCs components and variation of the corresponding functional group index.

**Figure 12 materials-14-00947-f012:**
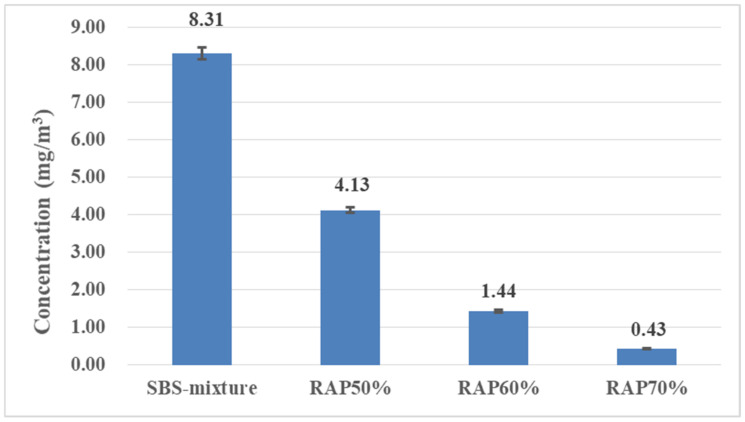
Comparison of VOCs emission of recycled asphalt mixture with a high percentage of RAP during the mixing stage.

**Figure 13 materials-14-00947-f013:**
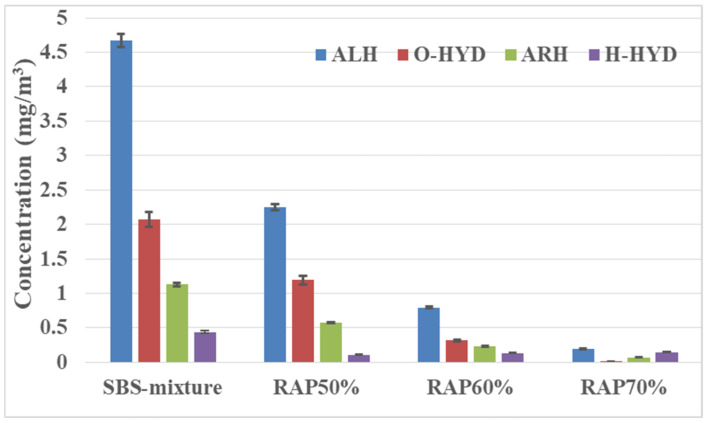
VOCs release in different categories of recycled asphalt mixture with a high percentage of RAP.

**Table 1 materials-14-00947-t001:** Fundamental properties for SBS modified asphalt and reclaimed asphalt pavement (RAP) binder.

Binder Type	Penetration 25 °C (0.1 mm)	Softening Point (°C)	Ductility 50 mm/min, 5 °C (cm)
SBS asphalt	68	56	48.6
RAP binder	45	73	18.7
Test Method	ASTM D5	ASTM D36	ASTM D113

**Table 2 materials-14-00947-t002:** Servicing performance of recycled asphalt mixtures.

RAP Ratio (%)	Porosity (%)	Residual Stability (%)	Tensile Strength Ratio (%)	Dynamic Stability (Frequency/mm)	Bending Strength (MPa)
50	4.01	94.6	93.4	4228	15.5
60	4.08	93.2	89.3	4437	13.4
70	3.91	89.4	84.2	4701	11.2
Specification requirements	≤5%	≥85	≥80	≥2400	-

## Data Availability

Data available in a publicly accessible repository.

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
