# Peer review of "Environmental Impact on VOCs Emission of a Recycled Asphalt Mixture with a High Percentage of RAP"

_materials, 2021, doi:10.3390/ma14040947_

Round 1

Reviewer 1 Report

The article entitled “Environmental impact on VOCs emission of recycled asphalt mixture with high percentage of reclaimed asphalt pavement (RAP)” is an interesting analysis of volatile compounds emissions. This is an issue that still attracts the attention of many scientists. Nevertheless, in the article there are several inaccuracies that require significant correction and supplementation. Their sample list is below:
1. The use of abbreviations, eg RAP in the title is not correct. This acronym has been well known for many decades and can be used in the body of the article.
2. Why the authors used only one reclaimed pavement material, which contains steel slag as aggregate. I am of the opinion that such a sample cannot be considered as representative. To make the analysis more reliable more RAP samples should be taken into account.
3. What filler was used to manufacture the asphalt mastic. This is of great importance due to the aging rate and the amount of volatile compounds emission. More information on this subject should be add to this article.
4. Did the authors define the LVE range prior to rheological testing? It is not known what range or value of strain they took into account.
5. What kind of technique did the authors use to measure the aging index values (FTIR)? Whether it was an ATR, KBr or other technique?
6. Are the authors invented equations of 1-4? If not, please provide the literature from which this information was obtained.
7. What was the precision of the measured functional group indices based on the authors' research?
8. What was “Standard test Methods of Bitumen….”? Can the authors provide a reference to the literature.
9. According to the reviewer, the Marshall test is not a representative test dedicated for mineral mixtures and has not been used by well known research centers for over a decade. In addition, Marshall stability test shows a weak correlation with the rheological properties of asphalt mixtures.
10. Fig. 4 and Fig. 5 should be presented as a Black curve, especially since the polymer modified bitumen was used in tests. 11. There is no information in the paper on mineral mix composition used for the results validation. Consequently, it is difficult to determine the importance of these studies. Furthermore, there are no standards to which authors referred to. Overall, the article still requires a lot of attention to be credible.

Author Response

Reviewer #1:

  1. The use of abbreviations, eg RAP in the title is not correct. This acronym has been well known for many decades and can be used in the body of the article.

       Thanks for your kind comments. The expression of title has been revised             and the files have been uploaded. 

  1. Why the authors used only one reclaimed pavement material, which contains steel slag as aggregate. I am of the opinion that such a sample cannot be considered as representative. To make the analysis more reliable more RAP samples should be taken into account. 

Thanks for your kind comments. The main purpose of our research is the  reuse of waste resources and relative environmental impact assessment. It is reported that Wu-Huang expressway is the earliest one to use steel slag in China [1] and the collected RAP sample can better reflect the aging condition of asphalt concrete under the function of environment and load. The impact of RAP on VOCs emission still needs further exploration and other kinds of RAP will be considered in our future work to improve the results.

  1. What filler was used to manufacture the asphalt mastic. This is of great importance due to the aging rate and the amount of volatile compounds emission. More information on this subject should be add to this article.

Thanks for your kind comments and apologize for the controversial expression. Considering the effect of chemical structure on VOCs emission, the comparison between SBS asphalt binder and extracted RAP asphalt binder was conducted without the addition of filler. In the preparation process of RAP asphalt binder, the mineral filler has been removed by centrifuge. The description as “asphalt mastic” has been revised to “asphalt binder” in the paper.

  1. Did the authors define the LVE range prior to rheological testing? It is not known what range or value of strain they took into account. 

Thanks for your kind comments. Details of the test process and parameter settings has been supplied in the uploaded file as follow:

“The strain is 2.0%.”

  1. What kind of technique did the authors use to measure the aging index values (FTIR)? Whether it was an ATR, KBr or other technique?

Thanks for your kind comments. Fourier transform infrared spectroscopy (FTIR) instrument was conducted to test the chemical functional groups of asphalt mastic. The asphalt film was used for the test which is prepared by dissolving 0.1g sample in 2ml CS2 and then dropping the solution onto KBr wafer until it volatilizes completely. The detailed information has been supplied in line 125-128.

  1. Are the authors invented equations of 1-4? If not, please provide the literature from which this information was obtained.

Thanks for your kind comments. The functional and structural indexes calculated from the band areas with FTIR results are conventional methods which have been mentioned in substantial researches. Relevant references have been added to the article.

[27] P.D.V.M.J.K. J. Lamontagnea, Comparison by Fourier transform infrared (FTIR) spectroscopy of different ageing techniques: application to road bitumens, Fuel 80 (2001) 483-488.

[28] Y. Li, S. Wu, Q. Liu, Y. Dai, C. Li, H. Li, S. Nie, W. Song, Aging degradation of asphalt binder by narrow-band UV radiations with a range of dominant wavelengths, Construction and Building Materials 220 (2019) 637-650.

[29] X. Zhao, S. Wang, Q. Wang, H. Yao, Rheological and structural evolution of SBS modified asphalts under natural weathering, Fuel 184 (2016) 242-247.

  1. What was the precision of the measured functional group indices based on the authors' research?

Thanks for your kind comments. The aliphatic functional group index ranges from 0.037-0.050; the aromatic functional group index ranges from 0.021-0.030; the carbonyl functional group index ranges from 0.001-0.006. The values have been marked in Figure 11.

  1. What was “Standard test Methods of Bitumen….”? Can the authors provide a reference to the literature. 

Thanks for your kind comments and the corresponding reference of “Standard test methods of bitumen and bituminous mixture for highway engineering” has been supplied in the uploaded file.

[30] Ministry of Transport of the People’s Republic of China. Standard test methods of bitumen and bituminous mixtures for highway engineering. JTG E20, 2011 (in Chinese).

  1. According to the reviewer, the Marshall test is not a representative test dedicated for mineral mixtures and has not been used by well known research centers for over a decade. In addition, Marshall stability test shows a weak correlation with the rheological properties of asphalt mixtures.

Thanks for your kind comments. The main purpose of this paper is the assessment of VOCs emission of RAP. Marshall test is widely used in the quality inspection of asphalt mixture. In China's specifications, the residual stability, dynamic stability and other indicators are used to verify whether the road servicing performance of asphalt mixture is qualified or not with high percentage of RAP.

  1. 4 and Fig. 5 should be presented as a Black curve, especially since the polymer modified bitumen was used in tests.

Thanks for your kind comments. The pictures have been updated in the revised version.

  1. There is no information in the paper on mineral mix composition used for the results validation. Consequently, it is difficult to determine the importance of these studies. Furthermore, there are no standards to which authors referred to. Overall, the article still requires a lot of attention to be credible.

Thanks for your kind comments. The grading curve of AC-13 asphalt mixture with different RAP ratio has been supplied in the uploaded file.

Reviewer 2 Report

This paper studies the environmental impact on VOCs emission of recycled asphalt mixture with high percentage of RAP.

The experimental program is extensive and detailed and the quality of the results’ discussion in its current form satisfies the requirements necessary for a research paper.

Some minor revisions:

  • figures 4 and 5: please improve the quality and the legibility, in terms of colours and text dimensions
  • figure 9: delete symbol of percentage from columns, in order to improve its legibility
  • introduction: please move the references between square brackets at the end of the sentence.
  • to improve the Scientific Soundness of the paper clarifying better the VOCs origin and the related problems.
  • For the introduction please see: A laboratory and filed evaluation of Cold Recycled Mixture for base layer entirely made with Reclaimed Asphalt Pavement. Construction & Building Materials. Scopus Code: 2-s2.0-85012293798

Author Response

Reviewer #2:

  1. figures 4 and 5: please improve the quality and the legibility, in terms of colours and text dimensions

Thanks for your kind comments. The pictures have been updated in the revised version.

  1. figure 9: delete symbol of percentage from columns, in order to improve its legibility

Thanks for your kind comments. The pictures have been updated in the revised version.

  1. introduction: please move the references between square brackets at the end of the sentence. to improve the Scientific Soundness of the paper clarifying better the VOCs origin and the related problems.For the introduction please see: A laboratory and filed evaluation of Cold Recycled Mixture for base layer entirely made with Reclaimed Asphalt Pavement. Construction & Building Materials. Scopus Code: 2-s2.0-85012293798

Thanks for your kind comments. The format of references in the introduction has been updated in the uploaded file.

Reviewer 3 Report

Dear Authors,

in my opinion, your manuscript entitled: "Environmental impact on VOCs emission of recycled asphalt mixture with high percentage of reclaimed asphalt pavement (RAP)" can be published in Materials journal after major revision.

The main topic of your study is the investigation of quantify the VOCs emission with high percentage of RAP in recycled asphalt mixture. You showed the the correlation between the chemical composition of recycled asphalt mastic and VOCs emission. This study shows that the potential VOCs reduction effect of the application with recycled asphalt mixture.

After the careful review of your manuscript, I have several remarks and comments which you can find below.

Can you add the information about the pre-treatment of plastic bag which was used to collect of gases during the analysis? It was purred using inert gas or atmospheric air was removed?

Can you add the information about the stirring speed during in the mixing of asphalt samples.?

How you can explain that when the RAP mastic content reaches 80% (R-80% sample), the phase angle of the mixed asphalt mastic decreases obviously and gradually approaches that of the extracted RAP mastic and the value of phase angle is stable around 80o in the temperature range between 70 and 80 oC?

Can you present of the areas of FTIR bands which are seen in Figure 7 and you have to compare the obtained values for selected samples.

You show the VOCs emission concentration of recycled asphalt mastic with different mixing ratio in Figure 8. You have to show the errors of the obtained values of VOCs concentration. You have to show also the errors for obtained values of the composition distribution of VOCs in recycled asphalt binder with different mixing ratio which were shown in Figure 9.

In Figure 12, you present the emission in different categories of VOCs from the recycled asphalt mixture with high percentage of RAP. You have to show the errors for obtained values of concentration.

Kind regards,

Reviewer

Author Response

Reviewer #3:

  1. Can you add the information about the pre-treatment of plastic bag which was used to collect of gases during the analysis? It was purred using inert gas or atmospheric air was removed?

Thanks for your kind comments. Before starting to collect gas sample, it is necessary to use the high purity nitrogen for replacement of gas sampler bag that is filling the gas and then letting it go, and repeating it five times. When last replacement is finished, the sampling pump is used to empty the gas in the sampling bag which is waiting for the sample collection. The detailed information has been supplied in the revised file.

  1. Can you add the information about the stirring speed during in the mixing of asphalt samples.?

Thanks for your kind comments. It began to collect released gas sample when the mixed asphalt binder was stirred at a constant rate of 300 r/min for 10 min to ensure the homogeneity of the mixture. The detailed information has been supplied in the revised file.

  1. How you can explain that when the RAP mastic content reaches 80% (R-80% sample), the phase angle of the mixed asphalt mastic decreases obviously and gradually approaches that of the extracted RAP mastic and the value of phase angle is stable around 80o in the temperature range between 70 and 80 oC?

Thanks for your kind comments. When the RAP binder content is less than 80%, the RAP binder present better mixing effect with SBS virgin asphalt binder under the regeneration effect of asphalt rejuvenator agent. However, the regeneration effect is not enough to ensure the mixing effect with the two kinds of asphalt binder. The plateau region of phase angle at 70-80 ℃ for R-80% indicates the transition from high elastic state to viscous flow state. Compared with other mixed asphalt binder, R-80% sample contains less viscous components [1].

[1] Cao, Z.; Huang, X.; Yu, J.; Han, X.; Wang, R.; Li, Y., Study on all-components regeneration of ultraviolet aged SBS modified asphalt for high-performance recycling. Journal of Cleaner Production 2020, 276, 123376.

  1. Can you present of the areas of FTIR bands which are seen in Figure 7 and you have to compare the obtained values for selected samples.

Thanks for your kind comments. In fact, it is not suitable to simply compare the difference of each FTIR band area because asphalt film for the test does not have a constant thickness and the spectrum is not at the same absorbent series. Therefore, the general method at present to assess the chemical structure change of asphalt binder at different condition is to use the functional group indexes (as mentioned in 2.4).  The calculation indexes can avoid using spectrum normalization which has not been taken into practice.

  1. You show the VOCs emission concentration of recycled asphalt mastic with different mixing ratio in Figure 8. You have to show the errors of the obtained values of VOCs concentration. You have to show also the errors for obtained values of the composition distribution of VOCs in recycled asphalt binder with different mixing ratio which were shown in Figure 9.

Thanks for your kind comments. The pictures have been updated in the revised version.

  1. In Figure 12, you present the emission in different categories of VOCs from the recycled asphalt mixture with high percentage of RAP. You have to show the errors for obtained values of concentration.

Thanks for your kind comments. The pictures have been updated in the revised version.

Round 2

Reviewer 1 Report

A reviewer's opinion the Authors made an effort to correct major remarks satisfactory.

Reviewer 3 Report

Dear Authors,

in my opinion, your corrected manuscript entitled: "Environmental impact on VOCs emission of recycled asphalt mixture with high percentage of reclaimed asphalt pavement (RAP)" can be published in Materials journal in the present form.

Kind regards,

Reviewer